# Novel Procedure for Automatic Registration between Cone-Beam Computed Tomography and Intraoral Scan Data Supported with 3D Segmentation

**DOI:** 10.3390/bioengineering10111326

**Published:** 2023-11-17

**Authors:** Yoon-Ji Kim, Jang-Hoon Ahn, Hyun-Kyo Lim, Thong Phi Nguyen, Nayansi Jha, Ami Kim, Jonghun Yoon

**Affiliations:** 1Department of Orthodontics, Asan Medical Center, University of Ulsan College of Medicine, Seoul 05505, Republic of Korea; yn0331@ulsan.ac.kr (Y.-J.K.); nayansijha@ulsan.ac.kr (N.J.); 2Department of Orthodontics, Chungang University Gwangmyeong Hospital, Gwangmyeong 14353, Republic of Korea; ajh0225@cauhs.or.kr; 3Department of Mechanical Design Engineering, Hanyang University, Seoul 04763, Republic of Korea; dlagusry9606@hanyang.ac.kr (H.-K.L.); npthong2511@hanyang.ac.kr (T.P.N.); 4BK21 FOUR ERICA-ACE Center, Hanyang University, Ansan 15588, Republic of Korea; 5Seoul Ami Orthodontic Private Practice, Incheon 22011, Republic of Korea; cadenmkim@gmail.com; 6Department of Mechanical Engineering, Hanyang University, Ansan 15588, Republic of Korea

**Keywords:** orthodontics, cone-beam computed tomography, intraoral scan, 3D registration

## Abstract

In contemporary practice, intraoral scans and cone-beam computed tomography (CBCT) are widely adopted techniques for tooth localization and the acquisition of comprehensive three-dimensional models. Despite their utility, each dataset presents inherent merits and limitations, prompting the pursuit of an amalgamated solution for optimization. Thus, this research introduces a novel 3D registration approach aimed at harmonizing these distinct datasets to offer a holistic perspective. In the pre-processing phase, a retrained Mask-RCNN is deployed on both sagittal and panoramic projections to partition upper and lower teeth from the encompassing CBCT raw data. Simultaneously, a chromatic classification model is proposed for segregating gingival tissue from tooth structures in intraoral scan data. Subsequently, the segregated datasets are aligned based on dental crowns, employing the robust RANSAC and ICP algorithms. To assess the proposed methodology’s efficacy, the Euclidean distance between corresponding points is statistically evaluated. Additionally, dental experts, including two orthodontists and an experienced general dentist, evaluate the clinical potential by measuring distances between landmarks on tooth surfaces. The computed error in corresponding point distances between intraoral scan data and CBCT data in the automatically registered datasets utilizing the proposed technique is quantified at 0.234 ± 0.019 mm, which is significantly below the 0.3 mm CBCT voxel size. Moreover, the average measurement discrepancy among expert-identified landmarks ranges from 0.368 to 1.079 mm, underscoring the promise of the proposed method.

## 1. Introduction

With the advancement of new Computer-Aided Design and Computer-Aided Manufacturing (CAD/CAM) technologies in dentistry, clinicians are increasingly employing virtual simulations for various dental treatments. Dental cone-beam computed tomography (CBCT) is a widely utilized imaging modality known for its low-dose radiation capabilities, enabling the visualization of teeth and craniofacial structures. It provides comprehensive 3D diagnostic capabilities and finds extensive application in various dental treatments, including orthodontics, dental implants, and orthognathic surgery.

For instance, the visualization of impacted teeth supports the precise formulation of treatment plans to align and establish final tooth positions for orthodontic treatment goals. In the case of dental implants, fixture diameter, and positioning can be determined based on a patient’s alveolar bone morphology. In orthognathic surgery, a 3D virtual model of patients’ jaws can be generated, enabling the computer-simulated surgery to accurately correct jaw deformities. Subsequently, leveraging the simulation data, various devices like clear aligners for orthodontic treatment, 3D printed surgical guides, and splints for dental implants and orthognathic surgery, respectively, can be manufactured.

However, it is essential to acknowledge that the spatial resolution of dental CBCT may be insufficient for precisely defining tooth crowns and occlusion. Therefore, the imperative task of acquiring and integrating high-resolution IOS data with CBCT data arises to overcome this limitation [1,2].

The integration between the IOS and CBCT data, referred to as registration, aligns these two data sets by utilizing common reference points captured separately via the IOS and CBCT machines. The accuracy of this registration process is of utmost importance, especially in the context of appliance manufacturing, as it directly influences the success of treatments following the translation of virtual simulations based on the registration results.

For example, even a small amount of variance during the registration process between CBCT and IOS data for the creation of surgical splints in orthognathic surgery can lead to inaccuracies in jaw positions during surgery, which can have permanent consequences [3]. While manual registration, supported by various commercial software [4,5,6,7], is a feasible option, it demands a significant investment of time and effort and is susceptible to the operator’s level of expertise [8,9]. 

The automated registration of heterogeneous CBCT and IOS data presents several challenges, despite their shared patient origin. CBCT comprises a range of cross-sectional images, including the skin, craniofacial skeleton, tooth crown, root, and soft tissues. The low-dose imaging modality often results in limited contrast between different anatomical structures, making object recognition difficult. In contrast, IOS provides high-resolution surface data of the tooth crown and adjacent soft tissues. To achieve accurate registration between CBCT and IOS data, an initial segmentation of the tooth crown from the CBCT data is essential.

Jang et al. [10] developed a method to generate metal-artifact-free panoramic images from CT scans, combined with a tooth detection approach that classified teeth into four types based on their morphology. However, identifying neighboring teeth in panoramic images, especially in cases involving missing teeth with similar types, remains a challenging task. Chang et al. [11] introduced a deep-learning-based metal artifact reduction method that utilized intra-oral scan data as supplementary information to aid tooth segmentation. Nonetheless, the training data were limited to axial views, which had limitations in achieving a clear separation between the upper and lower teeth in the 3D CBCT model.

To improve the alignment between CBCT and IOS data, particularly in the crown region, an initial alignment was attempted using principal component analysis (PCA) [12], a commonly used statistical method for dimensionality reduction [13]. PCA transforms data into a new coordinate system, effectively capturing data variation in fewer dimensions. However, conventional PCA can be sensitive to noise [14,15], making it challenging to differentiate noise from signal variance and handle missing observations.

In contrast, the random sample consensus (RANSAC) algorithm [16], a machine learning technique, estimates model parameters via iterative sampling, enabling the achievement of optimal fitting results in datasets containing both inliers and outliers [17]. 

In this paper, we not only implemented a Mask-RCNN [18], which was retrained for CBCT slices in the sagittal direction of the NHP posture, to facilitate the clear separation of upper and lower teeth from the entire CBCT raw data as the appropriate region of interest (ROI), but we also measured the position of the tooth arch in the sagittal view, supported by the tooth arch extraction scheme along the axial view [19]. This approach allowed us to separate and categorize the teeth into upper and lower teeth based on the panoramic view obtained using Mask-RCNN. Once the CBCT and IOS data were properly extracted, focusing on the crown part, we applied the RANSAC algorithm for the initial alignment. In the case of CBCT, it included both the crown and root, while IOS only encompassed the crown. After achieving the initial alignment, matching the data sets based on the center position of the crown, a fine alignment was performed using the point-to-plane algorithm [20].

To validate the integration of automatically registered data, we employed the Bland–Altman method [21,22,23], focusing on the cross-sectional tooth area. The quantitative integration assessment involved the chamfer and Hausdorff distances [24,25] between matched CBCT and IOS data planes. These distances were evaluated by three experienced orthodontists, who manually measured four key crown feature positions.

## 2. Data Preparation 

### 2.1. 3D Dental Data

This study was approved by the institutional review board of Hallym University (2022-04-018). We collected CBCT and plaster mold from 200 patients who had visited the Department of Orthodontics at Kangnam Sacred Heart Hospital, Hallym University, Seoul, Republic of Korea, for orthodontic diagnosis. IOS 3d data were collected from plaster molds of patients. Patient consent was waived due to the retrospective nature of this study. 

A single CBCT with a full field of image measuring 230 × 170 (mm × mm) was acquired using an I-CAT CBCT scanning machine (KaVo Dental GmbH, Biberach, Germany). The operational parameters for this scan were set at 120 kV, 37.1 mA, with a voxel size of 0.3 mm and a scan time of 8.9 s. It consists of 576 slices at a resolution of 768 × 768 × 576 pixels with a slice thickness of 0.3 mm. The capture conditions were standardized with the patient’s eyes instructed to focus on a 400 mm × 500 mm mirror, which was positioned on the wall approximately 1500 mm away from the patient’s head (maintaining the natural head position, NHP [26]). Prior to operating the CBCT scanning machine, patients were asked to exercise their heads up and down in accordance with Solow and Tallgren’s method [26].

IOS data were acquired using the Trios3 intraoral scanner (3Shape, Copenhagen, Denmark). This scanner falls into the category of structured light scanners and employs confocal microscopy and ultra-optical scanning technologies. It offers a field of view measuring 20.6 mm × 17.28 mm with an accuracy of 6.9 ± 0.9 µm.

Figure 1 illustrates the CBCT and IOS systems, and Table 1 provides the specifications of the measuring machines used in this study.

### 2.2. ROI Extraction 

To enhance both the efficiency and accuracy of the automated registration process between cone-beam computed tomography (CBCT) and intraoral optical scanning (IOS) data, a necessity arises to selectively extract only the pertinent target objects, namely teeth, from the comprehensive dataset. Traditional CBCT datasets encompass not only the geometric attributes of teeth but also a composite depiction of the craniofacial skeleton, muscles, skin, and additional soft tissues. Similarly, IOS data inherently encompass the geometric profiles of both teeth and gingival tissues.

In the case of CBCT, the region of interest considered for this segmentation ranged from A-point to Pog-point along the *z*-axis and between two mandible heads with respect to the *y*-axis.

In the extracted CBCT image X, X(x,y,z). represents the Hounsfield units in the CBCT at the voxel position (x,y,z). Firstly, along the sagittal direction, the 3D segmentation process is performed by applying the re-trained Mask-RCNN [18] on every slice. 

Different from the previous research [10,11], which processes along the z direction of the inputted CBCT, we had difficulties dealing with cases in which patients clench their teeth. In this situation, if it is processed along the z direction of the inputted CBCT, there are multiple positions where there is no empty space between the upper and lower teeth and the boundary consequently cannot be defined. Therefore, in this research, the method for tooth segmentation is performed along the y direction, which allows us to observe the cross-sectional shape, including the crown and root section of a tooth, and is considered the most convenient circumstance for defining the boundary for upper and lower teeth.

In every slice s, the segmentated mask ${\;M_{sagittal}}_{n}(s)$ is represented by a center point $p_{{\;M_{sagittal}}_{n}(s)}$ of the detected bounding box $\left(x_{n}\left(s\right),z_{n}\left(s\right),h_{n}\left(s\right),w_{n}\left(s\right)\right)$, as shown in Figure 2. The coordinate of the representative point $p_{{\;M_{sagittal}}_{n}(s)}$ is shown in Equation (1).
(1)$$p_{{M_{sagittal}}_{n}\left(s\right)}\left(x,y,z\right)=\left(x_{n}\left(s\right)+\frac{w_{n}\left(s\right)}{2},s,z_{n}\left(s\right)+\frac{h_{n}\left(s\right)}{2}\right)$$

Secondly, on the axial view, the dental arch curve C is detected on the mean intensity projection using the technique proposed in the previous study of Ahn et al. [19]. Based on the obtained results of the dental curve, the panoramic image is extracted using by Equation (2).
(2)$$P_{X}\left(c,z\right)=\int_{-w}^{w}X\left(r\left(c\right)+tn\left(c\right),z\right)dt,$$
where c is the index of the point in the dental arch, $r\left(c\right)\in C$, n(c) is the unit normal vector at $r\left(c\right)$, and w is the considered range from the dental arch along the normal vector n(c), as shown in Figure 3. 

Thirdly, based on the obtained panoramic view, teeth segmentation is performed by applying the Mask-RCNN. The masks obtained are split into upper masks ${\;M_{pano}}_{upper}$ and lower mask ${\;M_{pano}}_{lower}$, which, respectively, represents upper and lower teeth, by applying the linear regression to estimate the boundary between the jaws [19], as shown in Figure 4a.

The representative points $p_{{\;M_{sagittal}}_{n}(s)}$, which represents segmentated masks along the sagittal axis, are projected on the panoramic image as $p_{pano}^{sagi}$, similarly to the converting process for panoramic view in Equation (3).
(3)$$p_{pano}^{sagi}=\left\{p\left(r\right)=\left(q\left(r\right),z\left(r\right)\right):1\leq r\leq N_{r}\right\},$$
where $N_{r}$ is the number of representative points projected.

Consequently, as shown in Figure 4b, considering cells ${px}_{{pano}_{upper}}\in {M_{pano}}_{upper}$, if $p\left(r\right)\in {px}_{{pano}_{upper}}$, the segmented mask ${M_{sagittal}}_{n}(s)$, which is represented as $p\left(r\right)$, is collected into the group of upper teeth masks ${\tilde{X}}_{upper}$. Similarly, if $p\left(r\right)\in {px}_{{pano}_{lower}}$, the segmented mask ${M_{sagittal}}_{n}(s)$ is given into the group of lower teeth ${\tilde{X}}_{lower}$, with the obtained results shown in Figure 4c.

Concerning the IOS data, with the objective of optimizing the efficacy of the matching procedure, exclusive focus is placed on the segment corresponding to dental structures. This selected dental portion is then employed for alignment with the CBCT data. Consequently, the need arises for a methodology to segregate the dental structures encompassing teeth and gums from the original IOS dataset. The color attributes inherent in the IOS point cloud are harnessed to categorize each individual point within the dataset into one of two distinct groups. Within the input IOS point cloud, every point is characterized by RGB color values, denoting the red, green, and blue color channels. To enhance user-friendliness, these values are subsequently converted into the Hue–Saturation–Value (HSV) color model, where the Hue channel specifically encapsulates color information. In this proposed framework, the K-Nearest Neighbor (KNN) machine learning model [27] is enlisted for the purpose of categorizing each individual point within the IOS dataset into two primary color groups: one representing teeth and the other representing gums. In effectuating this classification, the KNN model assesses neighboring points that lie within a pre-defined distance based on the H, S, and V values of each given point. The predominant color group among these proximate points is then assigned to the color category of the particular input point under consideration, as illustrated in Figure 5. 

In some cases, CBCT data and IOS data did not align due to patients having orthodontic appliances or completed treatment. For these patients, we utilized their initial plaster models obtained at the outset of treatment, coinciding with CBCT data acquisition, for this study. These plaster models were scanned to obtain IOS models via the process depicted in Figure 6. Since some of these IOS data had a fixed white color, manual separation of teeth and gum was performed within the Trios 3 program as an alternative to the color-based separation method proposed in this study.

## 3. Registration Algorithm

Regarding the data derived from CBCT and IOS sources, both manifested as point clouds, an initial step involves the application of voxel down-sampling. This reduction in point cloud density serves a dual purpose: diminishing computational workload and refining registration precision. The automated registration procedure unfolds systematically, encompassing two distinct phases of alignment: a preliminary coarse alignment and a subsequent refined alignment. This division is grounded in the specific aim of achieving alignment between the two dataset sets, culminating in an origin-based correspondence. Notably, the coarse alignment phase is intentionally engineered to avoid extensive iterations, thereby bolstering registration efficiency. The subsequent refined alignment stage consummates the automated registration process, affording heightened precision via the implementation of a localized matching algorithm. 

Initially, the RANSAC algorithm is employed to achieve a rough alignment of the two models’ positions during the initial matching process. The RANSAC algorithm consists of two main stages: the hypothesis process and the verification process. In the hypothesis process, a sample model is created by randomly selecting a subset of source data. The verification process then compares this model with the target data, recording the count of matches. This iterative procedure selects matched data points with match rates surpassing a predefined threshold as the output, as shown in Figure 7.

After achieving the initial approximate alignment, the focus shifts to fine alignment, aiming to attain a model sufficiently precise for clinical utilization. The fine alignment process employs the iterative closest point (ICP) algorithm, which minimizes point-to-point distances to narrow the gap between the objects. In particular, we employ the point-to-plane variant well suited for three-dimensional alignment. Figure 8a,b depict the principle of the point-to-plane algorithm and the alignment process. A virtual plane is created within the target data points, and the source data points adjust along the direction of their corresponding plane’s normal vector. This iterative process effectively brings the datasets into close proximity, minimizing interpoint distances.

## 4. Validation

Following automatic alignment, the alignment’s efficacy is assessed via a triad of evaluation methodologies—statistical, quantitative, and clinical. Bland–Altman analysis, a statistical tool, probes the alignment trend between disparate datasets. For gauging alignment accuracy, we computed the 3D Euclidean distances’ mean and standard deviation, which provided a measure of alignment precision. Lastly, to gauge the aligned data’s clinical viability, dentists employed passive evaluation methods.

### 4.1. Statistical Validation

Bland–Altman analysis, extensively applied in medical, healthcare, and chemical domains, is adopted here to assess the matching trend between the two datasets. The graph plots the difference and mean difference as axes, constructing a 95% confidence interval. As the datasets vary in three-dimensional coverage, the aligned data’s crown segment is coronally sliced, and cross-sectional areas in this plane are analyzed. Figure 9 showcases the Bland–Altman results for six randomly selected cases. The presented outcomes indicate an absence of a pronounced trend, with over 95% of the 20 points falling within the confidence interval. This suggests a propensity for the two datasets to align.

### 4.2. Quantitative Validation

To assess matching accuracy, the mean and standard deviation of 3D Euclidean distances are calculated for quantifying alignment precision. A color-mapping scheme was implemented to visualize inter-dataset distance discrepancies. Colors ranged from blue (0 mm minimum) to red (maximum). Figure 10 illustrates this technique: Figure 10a depicts the lingual perspective of the same teeth, while Figure 10b presents a distance histogram. The red dotted line within the histogram signifies the average distance, incorporating standard deviation and Fitness/Inlier_rmse. Table 2 outlines registration accuracy for a specific set of 10 randomly chosen cases. Remarkably, the mean distance is 0.234 mm, with a standard deviation near 0.132 mm.

### 4.3. Clinical Validation

To gauge the practicality of automatically registered data in clinical contexts, dentists employ passive evaluation methods utilizing aligned data. Figure 11a,b demarcate specific parameter locations for assessment. Figure 11a segments human teeth into four sections, utilizing ball cuff, U6MB, L6MB, and C measurements as parameters.

Figure 11c outlines the clinical evaluation process involving these parameters. It encompasses documenting parameter positions in each organized dataset and noting corresponding parameter distances. This dual-weekly assessment involves three orthodontists, ensuring reliability by averaging expert outcomes. Euclidean distances between parameter three-dimensional coordinates are measured collectively via the ‘MeshLab’ program, a commercial software.

Manual measurement results for the same model by clinical experts are shown in Figure 12. The outcomes are visually depicted in Figure 12. The assessments, performed by three distinct specialists across the same ten cases, are presented in the form of a box plot. The ultimate findings portray the average measurements obtained from these evaluations. The potential clinical applicability of the research methodology was gauged via the insights gleaned from the specialists’ evaluations.

## 5. Discussion

In the field of dentistry, the progression of CAD/CAM technologies has transformed the clinical landscape by replacing traditional alginate and rubber impression materials with intraoral scanners. Moreover, the adoption of low-dose CBCT imaging has enabled a cost-effective visualization of bone and teeth. The integration of computer simulation techniques has revolutionized treatment planning and the fabrication of dental appliances and prostheses, ushering in a digital era marked by the replacement of manual laboratory procedures with digital milling and 3D printing.

Despite the rapid technological advancements, certain limitations persist in terms of accuracy. Analogous to conventional diagnostic and treatment methods, errors have been reported in the use of digital tools. These errors may occur throughout the clinical pathway, including data acquisition, curation, and appliance fabrication. These limitations are particularly apparent when attempting to utilize various types of digital data to construct a comprehensive 3D model of a patient’s dental and craniofacial structures.

While CBCT provides images of bones and teeth, the resolution falls short for utilizing images of tooth crowns in appliance fabrication. Although efforts have been made to segment teeth from CBCT, the accuracy remains insufficient for clinical applications [10,28,29]. As a result, a critical necessity emerges for the registration of intraoral scan data and high-precision surface data with CBCT in clinical applications like the development of surgical guides and orthodontic clear aligners. This integration process, known as registration, relies on a common reference point established using the respective instruments. While the manual alignment of these two data types is possible [4,5,6,7], it presents challenges in terms of being labor-intensive and time-consuming. Furthermore, its accuracy depends on the operator’s proficiency [8,9], leading to low inter-examiner reliability. Hence, the quest for automated registration of Intraoral Scan (IOS) and CBCT data becomes of paramount importance. The unique characteristics of IOS and CBCT data call for customized data acquisition strategies that align with the restricted Region of Interest (ROI) defined by IOS.

Noh et al. [30] introduced a method for dental registration utilizing the iterative closest point algorithm, encompassing the matching of three distinct registration areas: buccal surfaces, lingual surfaces, and a combination of both. However, their approach lacked a procedure for the extraction of teeth models from the cone beam computed tomography (CBCT) data. Additionally, the research omitted details regarding the removal of gingival margin areas, leaving room for improvement in this aspect of the methodology. In contrast, Park et al. [31] presented a study employing a manual registration function and a point-based registration function to align CBCT and intraoral scanning (IOS) data. Their method required user interaction to select point pairs for matching. It is worth noting that the data used in their research were derived from an artificial skull model, which may not perfectly represent real patient data due to inherent differences. An intriguing approach was proposed by Deferm et al. [32], who introduced a novel soft tissue-based method for registering intraoral scans with CBCT scans. Their study commenced with the alignment of dentate jaws via the registration of the palatal mucosal surface, which was followed by a meticulous evaluation of accuracy at the individual tooth level. Subsequently, their unique methodology extended to the registration of fully edentulous jaws, which incorporated both the palatal and alveolar crest mucosal surfaces. However, the specifics of the iterative closest point (ICP) algorithm utilized in their research were drawn from commercial software, and the technical intricacies were not thoroughly elucidated. Piao et al. [8] contributed to the field by comparing multiple registration methods, including deep-learning-based registration, manual registration, surface-based registration, and point-based registration. These methods were integrated into commercial software packages employed in their research. As a result, this study emphasizes the utilization of these techniques, but detailed insights into their methodologies were not the central focus. Yang et al. [33] use a digital approach based on a single CBCT scan to transfer virtual intraoral scans to a physical mechanical articulator. This eliminates traditional procedures, streamlines workflows, and reduces chairside adjustments. The technique enables accurate intraoral scan mounting and virtual articulator parameter setting in prosthetic dentistry. However, it will require an external physical mechanical articulator, compared to the other computational techniques. Hamilton et al. [34] compare the registrations between IOS and the multiple-sized FOV of CBCT datasets. In their research, there is an observed increase in precision errors during intra-oral scan registration. Nevertheless, it is noteworthy that when an adequate number of well-distributed teeth are discernible within the small FoV CBCT, the precision of digital intra-oral scan registration seems to fall within clinically acceptable limits. However, the registration process was performed manually by a trained investigator. For applying deep-learning methods to integrate the CBCT and IOS, Lee et al. [35] introduce a study aiming to assess the precision of integrated tooth models (ITMs) generated through deep learning, which involves the fusion of intraoral scans and cone-beam computed tomography (CBCT) scans. The primary focus was on the three-dimensional (3D) analysis of root position in the context of orthodontic treatment. Additionally, this study aimed to juxtapose the fabrication process of ITMs using deep learning against the conventional manual method. However, the 3D segmentation mentioned in this research was performed using a commercial software package.

In this study, we leverage a retrained Mask-RCNN model on sagittal CBCT slices within the NHP posture, addressing limitations identified in previous investigations, particularly concerning the challenge of missing teeth [10,11]. Our method distinctly segregates upper and lower teeth via customized ROIs. Furthermore, we determine the sagittal position of the dental arch by concurrently employing a tooth arch extraction technique in conjunction with axial views [21]. On the other hand, for IOS data, we effectively separate it from the gingiva using a color-based KNN algorithm. Subsequently, our alignment strategy focuses on the tooth crown, capitalizing on the unique attributes derived from the extracted CBCT and IOS data. Notably, during the initial alignment phase, we opted for the RANSAC algorithm over PCA [12,13,14,15,16]. This choice was grounded in RANSAC’s resilience against outliers and missing data, resulting in a more reliable alignment process.

It is important to note that the careful use of this loop strategy contributes to the precision of the alignment process, enhancing the accuracy of the matched data. However, it is acknowledged that the matching time may be comparatively extended due to the iterative nature of the loop. Despite the inherent trade-off between precision and efficiency, our approach prioritizes achieving accurate alignment outcomes. The deliberate trade-off of increased processing time for accuracy is strategically balanced to yield results that align with the stringent demands of clinical applications, establishing the basis for practical integration of the method within the realm of dental practice.

## 6. Conclusions

This paper introduces an AI-driven dental system that integrates automated data extraction and matching techniques for IOS and CBCT data, thereby enhancing the diagnosis and treatment planning processes in dentistry. In CBCT data, a method for the precise segmentation of tooth-contact slices is proposed. This involves applying a retrained Mask-RCNN to sagittal images in NHP posture to isolate teeth. Dental arch positioning is achieved via axial-view tooth projections, while panoramic views are extracted from dental arches. The Mask-RCNN is further employed to distinguish upper and lower jaws via panoramic views and masks. For IOS data, conversion to point cloud format is followed by HSV color model utilization. A color-based KNN approach is applied to segregate teeth and gingiva. Addressing differing dataset scopes, sequential RANSAC and ICP algorithms are employed for matching and prioritizing tooth crown alignment. For validation purposes, a comparative assessment was conducted between the performance of the proposed method and expert-driven procedures. The discerned deviation between the proposed method and manual measurements was determined to be within acceptable bounds, thereby endorsing the potential viability of this method for practical deployment in clinical settings.

## Figures and Tables

**Figure 1 bioengineering-10-01326-f001:**
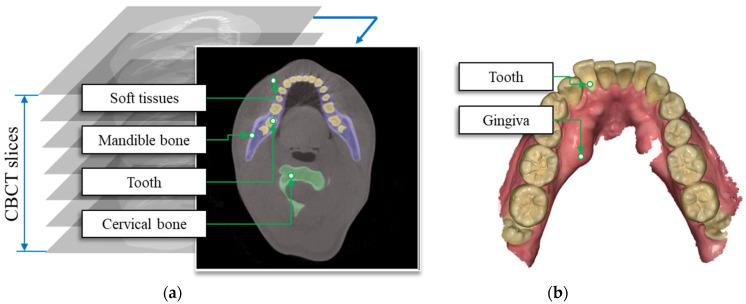
Characteristics of (**a**) cone-beam computed tomography and (**b**) IOS data.

**Figure 2 bioengineering-10-01326-f002:**
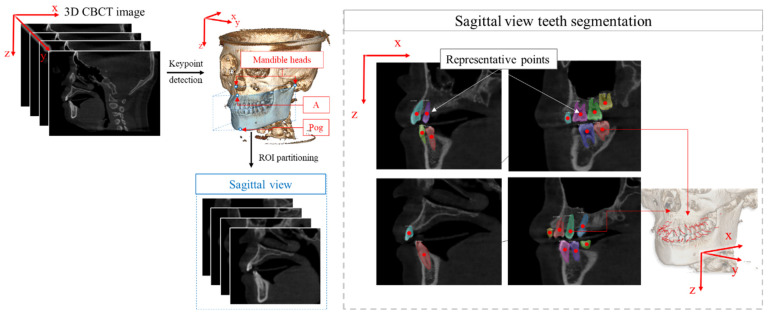
Teeth segmentation along sagittal views with Mask-RCNN.

**Figure 3 bioengineering-10-01326-f003:**
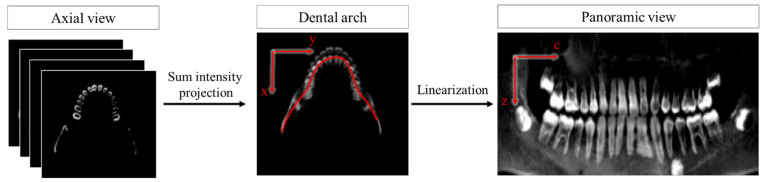
Panoramic view extraction based on the defined dental arch on the axial view.

**Figure 4 bioengineering-10-01326-f004:**
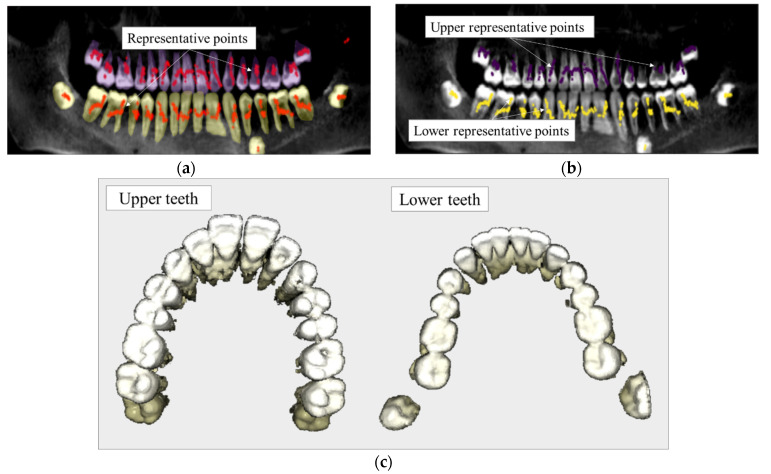
Combination of segmentation results. (**a**) Segmentation result on panoramic view for upper and lower teeth, (**b**) Separated upper and lower representative points, (**c**) 3D segmentation result for upper and lower teeth.

**Figure 5 bioengineering-10-01326-f005:**
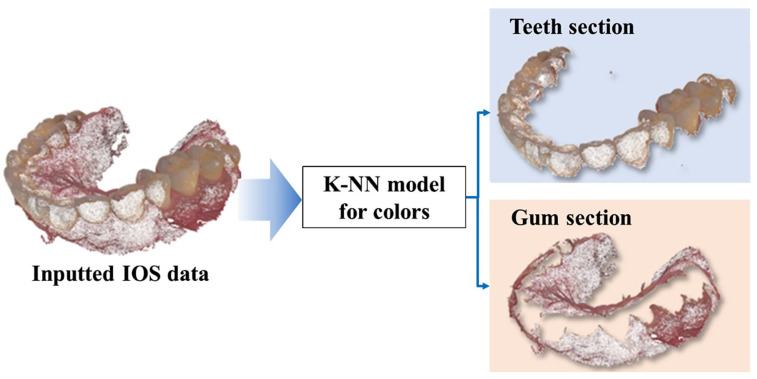
ROI extraction for teeth data from entire IOS data.

**Figure 6 bioengineering-10-01326-f006:**
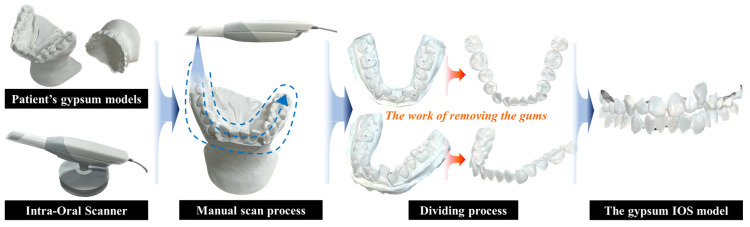
Process for requiring IOS data from gypsum model.

**Figure 7 bioengineering-10-01326-f007:**
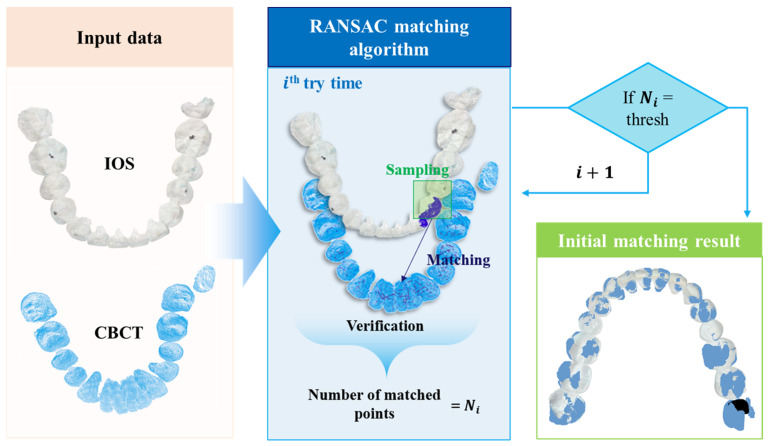
Application of RANSAC algorithm to rough alignment for two sets of teeth data.

**Figure 8 bioengineering-10-01326-f008:**
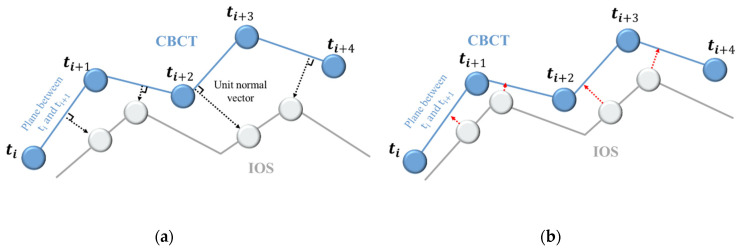
Concept of point-to-plane method for matching procedure between CBCT and IOS data: (**a**) Principle of point-to-plane method; (**b**) Source to access the plane created in the target.

**Figure 9 bioengineering-10-01326-f009:**
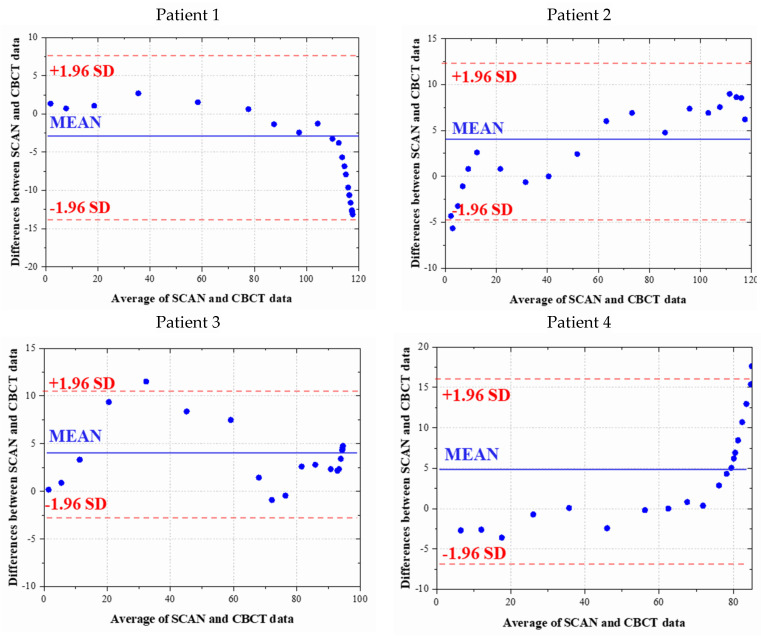
Results of Bland–Altman method: Data on 6 randomly selected patients.

**Figure 10 bioengineering-10-01326-f010:**
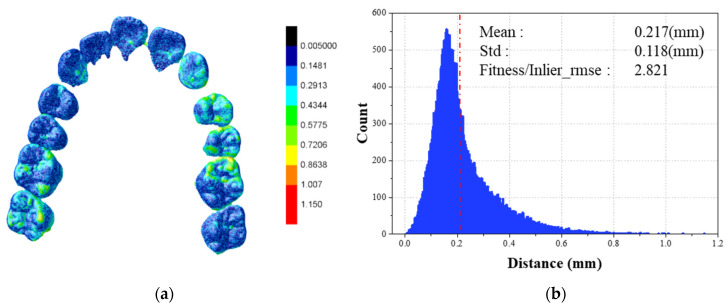
Error map of registration result from lingual view (**a**) and the histogram of distance between the corresponding points of two data (**b**).

**Figure 11 bioengineering-10-01326-f011:**
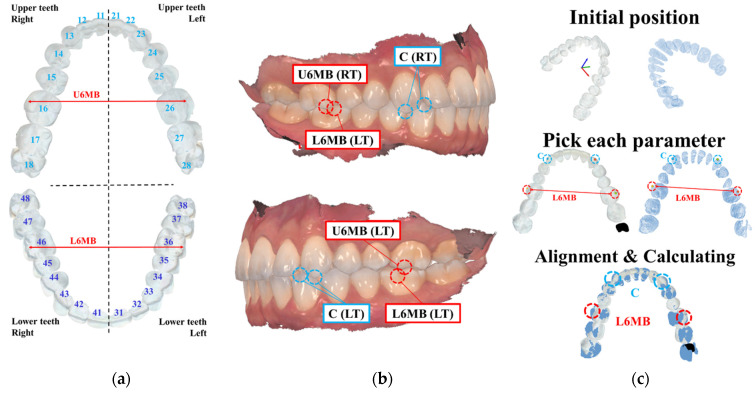
Position of features for evaluating consistency: (**a**) Number according to the position of the teeth and position of the teeth and positions of U6MB and L6MB; (**b**) Example of measurement parameter position of teeth; (**c**) Evaluation sequence on point cloud.

**Figure 12 bioengineering-10-01326-f012:**
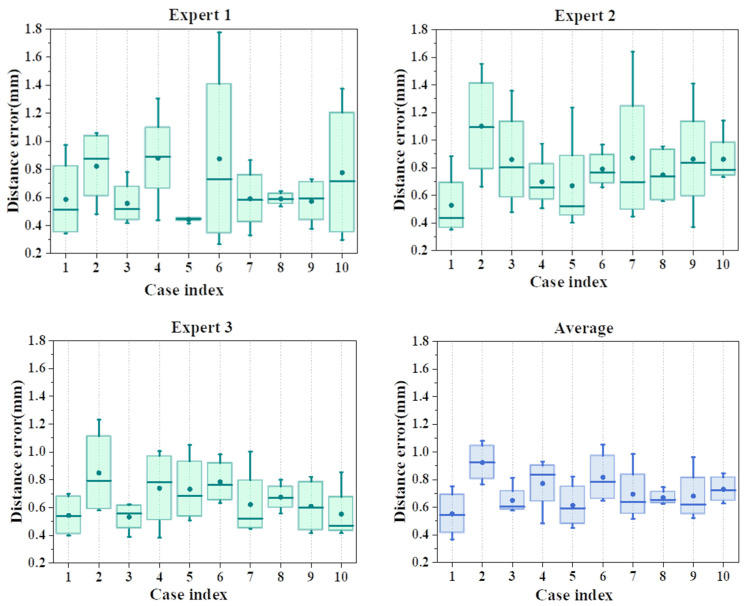
Distance between landmarks measured by experts (dentists) on aligned models.

**Table 1 bioengineering-10-01326-t001:** Specifications of digital dentistry device.

	CBCT	Intra-Oral Scanner (IOS)
Device brand	Imaging sciences international	3shape
Device model	Digital i-CAT FLX MV	Trios 3
Accuracy	0.3 mm (voxel size)	6.9±0.9 µm
Measuring time	~3 min/case	~5 min/case
Measurement area	Upper part of the neck	Teeth surface, gingiva

**Table 2 bioengineering-10-01326-t002:** Registration accuracy for 10 sample cases.

Value	Case Number										Average
	1	2	3	4	5	6	7	8	9	10	
Mean (mm)	0.226	0.292	0.215	0.221	0.217	0.229	0.249	0.227	0.217	0.249	0.234
Std (mm)	0.125	0.202	0.108	0.114	0.112	0.118	0.157	0.112	0.118	0.155	0.132
F/I	2.813	3.167	2.693	2.880	2.813	3.025	2.455	2.989	2.821	2.746	2.840

Std: standard deviation; F/I: Fitness/Inlier_rmse.

## Data Availability

Data sharing is unavailable due to ethical restrictions.

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
