# Peer review of "Novel Procedure for Automatic Registration between Cone-Beam Computed Tomography and Intraoral Scan Data Supported with 3D Segmentation"

_bioengineering, 2023, doi:10.3390/bioengineering10111326_

Round 1

Reviewer 1 Report

Comments and Suggestions for Authors

This manuscript is about: Novel procedure for automatic registration between CBCT and intraoral scan data with supported by 3D segmentation.

It's an interesting topic and introduces a novel 3D registration approach.

1. In the section of [Discussion], the abundance of this section seems not to be enough.

2.The format of [References] need to be corrected according the rules of the instructions to Authors.

Comments on the Quality of English Language

No comment

Author Response

We appreciate for the reviewer’s comments. In the revised manuscript, we have addressed all these points. Please, see the attachment.

Reviewer 2 Report

Comments and Suggestions for Authors

This study investigate an automatic data extraction and registration technique using CNN and AI. This will help dental clinician and technician to integrate digital technology into daily practice more efficiently. Therefore , it's a very interesting manuscript. There's one question: the IOS scan was based on stone models instead of real patients, Please clarify the IOS of stone model procedure, and add pictures of the data from this study.

Comments on the Quality of English Language

NA
